# Intralesional Infiltrations of Arteriosclerotic Tissue Cells-Free Filtrate Reproduce Vascular Pathology in Healthy Recipient Rats

**DOI:** 10.3390/ijms23031511

**Published:** 2022-01-28

**Authors:** Jorge Berlanga-Acosta, Maday Fernández-Mayola, Yssel Mendoza-Marí, Ariana García-Ojalvo, Indira Martinez-Jimenez, Nadia Rodriguez-Rodriguez, Raymond J. Playford, Osvaldo Reyes-Acosta, Laura Lopez-Marín, Gerardo Guillén-Nieto

**Affiliations:** 1Tissue Repair, Wound Healing and Cytoprotection Research Group, Biomedical Research Direction, Center for Genetic Engineering and Biotechnology, Ave. 31 S/N. e/158 and 190, Cubanacán, Playa, Havana 10600, Cuba; mfmayola@gmail.com (M.F.-M.); yssel.mendoza@cigb.edu.cu (Y.M.-M.); ariana.garcia@cigb.edu.cu (A.G.-O.); indira.martinez@cigb.edu.cu (I.M.-J.); nadia.rodriguez@cigb.edu.cu (N.R.-R.); osvaldo.reyes@cigb.edu.cu (O.R.-A.); gerardo.guillen@cigb.edu.cu (G.G.-N.); 2School of Biomedical Sciences, University of West London, St Marys Rd, Ealing, London W5 5RF, UK; ray.playford@uwl.ac.uk; 3Department of Pathology, Institute for Arteriosclerosis Research, Institute of Nephrology “Dr. Abelardo Buch”, Calle 26 y Línea del Ferrocarril, Vedado, Havana 10400, Cuba; laura.lopez@infomed.sld.cu

**Keywords:** critical limb ischemia, arterial disease, arteriosclerosis, angiopathy

## Abstract

Lower-extremity arterial disease is a major health problem with increasing prevalence, often leading to non-traumatic amputation, disability and mortality. The molecular mechanisms underpinning abnormal vascular wall remodeling are not fully understood. We hypothesized on the existence of a vascular tissue memory that may be transmitted through soluble signaling messengers, transferred from humans to healthy recipient animals, and consequently drive the recapitulation of arterial wall thickening and other vascular pathologies. We examined the effects of the intralesional infiltration for 6 days of arteriosclerotic popliteal artery-derived homogenates (100 µg of protein) into rats’ full-thickness wounds granulation tissue. Animals infiltrated with normal saline solution or healthy brachial arterial tissue homogenate obtained from traumatic amputation served as controls. The significant thickening of arteriolar walls was the constant outcome in two independent experiments for animals receiving arteriosclerotic tissue homogenates. This material induced other vascular morphological changes including an endothelial cell phenotypic reprogramming that mirrored the donor’s vascular histopathology. The immunohistochemical expression pattern of relevant vascular markers appeared to match between the human tissue and the corresponding recipient rats. These changes occurred within days of administration, and with no cross-species limitation. The identification of these “vascular disease drivers” may pave novel research avenues for atherosclerosis pathobiology.

## 1. Significance

This study describes the unprecedented observation that daily exposure of rats’ full-thickness wound granulation tissues to a homogenate of lower limb arteries of peripheral artery disease (PAD)/chronic-limb ischemia (CLI) affected patients, induced within a period of a week arteriolar wall thickening as well as other pathological microvascular changes in the recipient animals that very much mimic the integral vascular pathology inherent to the donors, and that are archetypical hallmarks of the human arteriosclerosis. We deem that this abnormal vascular remodeling and tissue expression changes detected in the rats’ wound arteries are mediated by the existence of soluble messengers/signaling factors that impose their pathologic programming over the otherwise normal angiogenic process. Of note, the action of these factors is not impaired by barriers of animal species. These observations incite us to consider that in addition to the well-known risk factors of arteriosclerosis, endogenous and unidentified drivers may be involved in the pathophysiology of systemic vascular disease. Furthermore, the identification of these actors may be tremendously useful for novel therapeutic interventions. This study supports and extends the notion on the existence of a transmissible pathologic vascular memory.

## 2. Introduction

PAD is a complex and multifaceted pathology with progressive dysfunctionality of both distributive and nutritional arterial circulation. This is an atherosclerotic obstructive process of lower extremity arterial vessels that impairs tissue perfusion and cellular respiratory capabilities [1,2,3]. Chronic limb-threatening ischemia represents the end stage of atherosclerotic PAD [4,5] inasmuch as diabetes mellitus remains as an important risk factor for atherosclerotic disease. Despite successful pharmacological interventions and educational programs PAD still contributes alarming limb amputation and early mortality figures [6,7,8].

The endothelium is the innermost single layer of a heterogeneous collection of cells that line the entire cardiovascular system. It receives and integrates information through a sophisticated sensory and signaling center that controls every cardiovascular function [9]. Thus, being the “hub cell” of the vascular tree, any endothelial dysfunction translates into a cascade of sequential vascular anomalies and pathological conditions that may ordinarily lead to arteriosclerosis [9,10,11].

Histological studies of diabetic ulcers granulation tissue (DUGT) conducted by our group raised the hypothesis of an existing “vascular tissue memory”. This notion is founded on the observation that within a period of a few days, DUGT has already reproduced a group of vascular defects including arteriolar walls thickening and luminal obliteration, all of which are expressed in distant, intact dermal vessels that take years-long to be implanted. Thus, although granulation tissue (GT) is a short half-life structure, evolutionarily programmed to fill an injury gap, it hosts endothelial cells that may be epigenetically instructed for the recapitulation of diabetic microangiopathy hallmarks [12]. More recently we described the unprecedented observation that diabetic impaired healing, and the archetypical histological markers of microangiopathy and neuropathy may be reproduced in healthy rats, through the administration of a cells-free filtrate (CFF) derived from diabetic ulcers granulation tissue, arteries, and nerves obtained from amputated lower limbs due to critical ischemia [13]. We consequently hypothesized that this inter-species pathologic phenotypic recapitulation is at least partially mediated by the existence of transmissible priming factors/soluble messengers, which represent the effects of the so-called diabetic metabolic memory.

Having observed that diabetic microangiopathy was consistently reproduced in healthy recipient rats in a manner reminding one of donor tissue histology, we decided to examine what changes could be elicited following the infiltration of an arterial CFF derived from non-diabetic patients afflicted by intractable lower limb ischemia. As described for diabetic tissues-derived CFF, exposure of rats’ granulation tissues to the arteriosclerotic donors’ homogenates, induced within a period of a week diverse vascular anomalies similar to those identified in the donors’ lower limb tissues. Although these are preliminary findings, the histological reproduction of vascular pathology in a healthy recipient animal was not anticipated, and appears to be not exclusive to diabetes and consequently, it is not influenced by a glucotoxic environment. On the basis of these preliminary evidences, further studies are warranted in order to fully examine the hypothetical existence of a transmissible “vascular tissue memory”.

## 3. Materials and Methods

### 3.1. Ethics and Consents

Use of human tissue was approved by hospital’s ethic committees and regulatory authorities in accordance to Helsinki Declaration. Subjects voluntarily gave informed written consent for the use of material surgically excised for clinical reasons. Popliteal artery and adjacent tissues fragments were obtained from two non-diabetic, atherosclerotic amputee subjects, at the National Institute of Angiology and Vascular Surgery (Arteriopathy Service), and Joaquin Albarrán Hospital. The brachial artery from a healthy donor was obtained at the Frank Pais National Orthopedic Hospital, both in the city of Havana, Cuba. Similarly, animal experiments were approved by the Animal Welfare Committee of the Center for Genetic Engineering and Biotechnology, Havana, Cuba after protocol review and discussion. Prevention of pain and distress in our experimental animals met all humane and methodological requisites, which in addition to being important for its own merits, was also relevant due to the facts that animals in distress have impaired healing processes [14,15].

### 3.2. Collection and Processing of Samples

#### 3.2.1. Collection of Human Samples

Popliteal arteries including the periadventitial tissues, skeletal muscle, smaller communicating vessels, and macroscopically normal cutaneous tissue samples were collected in the surgical act of lower extremity amputation of two donors. Popliteal artery, periadventitial tissues, and fragments of adjacent muscle were processed to obtain the CFFs used in this experiment. The first patient was a 61 years old non-diabetic, white subject, affected by chronic limb ischemia due to arteriosclerosis for about 10 years of evolution and classified as Fontaine IV/Rutherford category 5. The patient exhibited a well-delimited ischemic cutaneous plaque in the distal portion of the left leg which was not clinically infected. According to clinical records his glycemia levels ranged between 7–9 mmol/L. This tissue-derived homogenate was used for experiment 1. Additional material was processed as described and used for a second independent experiment. The second donor was a 70 years old non-diabetic, white female, with PAD-chronic limb ischemia for about 10 years and classified as Fontaine IV/Rutherford 5. The patient exhibited two well-delimited ischemic, non-clinically infected lesions in the external malleolar sector of the left leg. Glycemia levels averaged 8.1 mmol/L. The concurrent arterial control resided in a fresh brachial artery tissue collected from a male healthy donor (45 years old), following surgical amputation due to a traffic accident. Immediately following amputations in the surgical room, collected materials (arteries and adjacent soft tissues) were washed with sterile ice-cold normal saline to remove fibrin, blood and debris and cryopreserved in liquid nitrogen until processing for the CFFs preparation. Additional arterial and other tissues fragments including cutaneous and deep adjacent soft tissues were 10% buffered formalin fixed for histological analysis and characterization. The target artery and the adjacent vasculature were examined on the bases of broadly-accepted morphological parameters for arterial vascular pathology [16].

#### 3.2.2. Cell-Free Filtrate Preparation

Collected tissue samples were allowed to thaw, weighed and approx. 100 mg of wet tissue placed in a 2 mL vial containing 1 mL of normal saline, homogenized using a Tissue Lyser II for 3 min at 30 revolutions per second. Samples were then centrifuged at 10,000 rpm for 10 min at 4 °C, sterilized by filtration through 0.2 µm nitrocellulose filters (Sartorius Lab Instruments, Göttingen, Germany), aliquoted into sterile Eppendorf vials and stored at −70 °C. Prior to use, total proteins, glucose concentrations, and cytokine content of the samples were determined. Bicinchoninic Acid Protein Assay Kit (Sigma-Aldrich, St. Louis, MO, USA) and standard commercial ELISA kits for vascular cell adhesion molecule 1 (VCAM1), interleukin (IL)-1β and IL-6, (all from Abcam, Waltham, MA, USA) were used. For advanced glycation end products (AGE) concentration, a high sensitivity and specificity ELISA system was purchased from Wuxi Donglin Sci & Tech Development Co., Ltd. (Wuxi, China). According to the manufacturer, this kit is endowed with a detection range from less than 33.8 ng/mL to 8000 ng/mL. Vascular tissue levels of sirtuin-1 were determined by a sensitive ELISA commercial kit (Cloud-Clone Corp., Katy, TX, USA) having a lowest detection point of 0.29 ng/mL. Manufacturer’s instructions were followed for the analysis. The redox balance marker malondialdehyde (MDA) and nitrite/nitrate ratio were determined using commercial kits following the manufacturer’s instructions (both from Abcam, Waltham, MA, USA).

### 3.3. Induction of Skin Wound in Rats, Infiltration with Test Solutions and Subsequent Analyses

Young adult male Sprague Dawley rats (250–270 gr and *n* = 8 per group) were individually housed for 10 days of acclimatization in steel grid-bottomed cages, (to prevent contamination of wounds with bedding material) and allowed access to standard rodent chow and water ad libitum throughout the study. Following acclimatization and under anesthesia (ketamine (80 mg/kg)/xylazine 2% (10 mg/kg) cocktail) each rat underwent two dorsal, symmetrical, retro-scapular full-thickness wounds including *panniculus carnosum* with a 6 mm diameter disposable biotomes (Acu-Punch, Acuderm Inc., Fort Lauderdale, FL, USA) as previously described [17]. Immediately following wound induction and subsequent hemostasis, animals received local infiltration of test product (100 µg protein in 500 µL saline, or saline alone) into the wound (*n* = 16 wounds/group) on a once daily basis until end of experiment. The main goal of this study was to examine the local effects of the arterial tissue CFF derived from an arteriosclerosis obliterans patient. The rats were randomly assigned to three experimental groups which in addition to the main study group, two concomitant control groups were included. One based on the administration of healthy donor/normal artery tissue CFF, and a third one in which the rats received sterile normal saline which acted as a “clean” control before any rats’ tissue reaction to the administration of the human healthy material. Wound size on days 0, 3, 5 and 6 was determined by tracing wound margins on transparent polypropylene films, digitized, and two-dimensional digital planimetric calculation of wound closure determined. Results on day 0 (28 mm^2^) were defined as 100% [18,19]. Animals were autopsied on day 7 under terminal anesthesia to prevent complete wounds closure. The entire wound area with intact surrounding skin was excised, fixed in 10% buffered formalin, paraffin-embedded, and the 5-µm sections stained using H&E, Congo red, and Mallory’s trichrome for amyloid and collagen deposits identification, respectively. Images were captured using a BX43 Olympus microscope (Shinjuku City, Tokyo, Japan) coupled to a digital camera and central command unit (Olympus Dp-21), and processed using ImageJ software (ImageJ 1.48v, NIH, Bethesda, MD, USA) and H&E staining sections were used [20].

All histological assessments were performed by two independent, external pathologists under blinded conditions. Inflammatory infiltrate scoring (scored from 0–8), and fibro-vascular reaction (based on the degree of collagen bundle formation, orientation and maturity) were quantitated according to published methods [19,21]. The degree of vascular walls remodeling was quantitated from arterial wall-to-lumen ratios following described procedures [22,23]. The percentage of damaged nerve fibers, and degree of Wallerian degeneration was determined through validated methods [23,24]. Immunohistochemical study included a set of antibodies related to inflammation, oxidative stress, AGE/AGE receptor (RAGE) axis, atherosclerosis, vascular tone, and homeostatic control (Table 1). Briefly, tissue sections corresponding to each of the different arterial and periarterial samples were mounted on poly-l-lysine coated slides to reduce inter-tissue/experimental variations. The slides were dewaxed and rehydrated through graded washes of ethanol. Rehydrated slides were exposed to antigen retrieval solution for 20 min at 80 °C and washed with 0.05 M tris-buffered saline (pH 7.6) for 5 min. Endogenous peroxidase was blocked, and the tissue sections subsequently exposed to nonspecific binding blocking solution for 20 min. Antibody concentrations for paraffin-processed samples were used according to manufacturer’s specification. The immunolabeling reaction was developed as described in the Mouse and Rabbit Specific HRP/DAB (ABC) Detection IHC kit (Abcam, Waltham, MA, USA). In order to gain a more objective immunohistochemical assessment, the Galkowska technique was used [25] with minimal modification in the numerical scale. This procedure semiquantitatively grades the force of the immunoexpression by a simple scale: No staining: 0, weak staining: 1–3, moderate staining: 4–6, strong staining: 7–9. Given that our main examination target were arterial and periarterial structures of both human limb soft tissues, and rats granulation tissue, which *per se* includes arteries, veins, and extracellular matrix cells, we decided to consider the broadness of the immune reaction assigning a value of 1 point to each immunolabeled structure (for instance: extracellular matrix infiltrated foci of cells, arteriolar and venular walls, nerve fascicles, etc). Consequently, the Fedchenko and Reifenrath protocol [26] was used to calculate the final immunoreactive score (IRS) by the multiplication of the staining intensity per the number of reactive structures on each specimen. The data shown here represent the average of the blind scoring on serial sections of the same tissue sample (minimum of 4 readings for a human sample, and 14–16 for each rat group). As internal immunoreaction reference, a granulation tissue fragment of the donor’s diabetic-ischemic ulcer was used. Previous studies from our group had immunohistologically characterized this type of tissue [27]. Non-specific tissue labelling internal controls included the omission/replacement of the primary antibody by the background reducing antibody diluent (Abcam, Waltham, MA, USA), and normal rabbit serum (Boster Biological Technology, Pleasanton, CA, USA).

### 3.4. Statistical Analyses

Statistical processing was performed using GraphPad Prism software 6.01 (GraphPad Software, La Jolla, CA, USA). Statistical analyses were done through one way ANOVA followed by Holm-Sidak’s multiple comparisons test, or by the Kruskal-Wallis test followed by Dunn’s multiple comparisons test, for data with or without normal distribution, respectively. Immunohistochemistry data were analyzed using the unpaired t test or the Mann Whitney test, for data with or without normal distribution, respectively. AGE tissue expression analysis was performed using the Wilcoxon signed rank test. Data are expressed as mean ± SD and a *p* < 0.05 was established as statistically significant.

## 4. Results

### 4.1. Donors’ Arteries Histopathological and Biochemical Characterization

The histological examination of the healthy donor brachial artery confirmed the transmural preservation of the vascular tissue structure with no pathological changes (Figure 1A). On the contrary, the two patients’ popliteal artery serial fragments revealed typical changes of transmural sclerosis characterized by diffuse intimal thickening with areas of myofibroblastic proliferation and fibrotic induration, rupture of the internal elastic layer associated to media thickening, proliferative invasion of the intima and diffuse areas of fibrosis (Figure 1B). It was frequent to detect disperse foci of mononuclear cells infiltration within the thickened wall and the non-thrombotic occlusion of *vasa vasorum* due to endothelial hypertrophy (not shown).

An elemental biochemical characterization of the CFFs (Table 2) derived from the PAD samples, showed disparate pro-inflammatory cytokines, AGE, and MDA concentration values. However, common to both arteriosclerotic vessels-derived homogenate in relation to the healthy artery is that VCAM1 and sirtuin-1 concentrations appeared about 4-fold lower than the value observed in healthy donor sample. In correspondence with this, nitrite/nitrate ratio level of the healthy artery homogenate exceeded between 2.8 and 5-fold the measurements of the pathological samples. Aside from that and common to both arteriosclerotic donors we found an intrinsic angiogenic response based on de novo emerging vessels within the periadventitial tissues adjacent to the pathologic artery, skeletal muscle, and in deep cutaneous layers below the ischemic crust. Yet, this neovessels formation response was mostly abnormal with arteriolar walls thickening, an endothelial collar of bulky aspect, increased perivascular collagenization, and luminal occlusion (Figure 1C). The constellation of common vascular anomalies may be grouped and described as: (1) presence of malformed, or incompletely arranged vessels in nest-like formations embedded within a neoformed collagenous matrix (Figure 1D), (2) de novo-formed arterioles exhibiting a dense and thickened media layer with periadventitial collagen and subendothelial infiltrating hypercellularity (Figure 1E), (3) intima layer hyperplasia, often associated to fracture of *tunica elastica*; presence of fusiform, fibroblastic-like phenotype cells projecting bundles that encroach and obliterates the lumen (Figure 1F). The histological examination of the skeletal muscle fragments demonstrated the existence of an aberrant angiogenic response characterized by the intramyofibrillar onset of lumen-like structures often including the presence of an endothelial collar (Figure 1G).

### 4.2. Donors’ Homogenate Effect on Wound Healing Response, Nerves Integrity, and Inflammation

Experiments 1 and 2 showed that rats’ granulation tissue infiltrated with normal artery/healthy donor CFF did not exhibit wound contraction impairment, as compared to granulation tissue CFF samples from rats infiltrated with normal saline solution. The wounds of both groups had reduced their areas in more than 50%, exhibiting a normal fibrovascular response according to wound age for both experiments (Table 3). The impact of the PAD-derived CFF infiltration on the rate of wound closure was indifferent for the two experiments. Experiment 1: The CFF used in this experiment did not significantly modify wound contraction when compared with the two corresponding control groups. The percentage of degenerated nerve fibers was indifferent to the values calculated for the healthy artery. Both artery-derived CFF samples induced a significantly larger percentage of nerve degeneration when compared to the data from normal saline administration. The highest inflammatory score was calculated in those rats receiving PAD-derived CFF. Experiment 2: The percentage of wound closure for the group treated with donor-2 PAD homogenate was statistically similar to the one calculated for animals receiving saline solution (Table 3). This wound closure percentage was significantly lower than that induced by the healthy artery-derived CFF. PAD-derived CFF induced the largest percentage of damaged nerves with statistical significance as compared to normal saline treatment. Irrespective to the gap existing in mean values, no difference was shown when the PAD-derived CFF was compared to the effect of the derived from the healthy donor/normal artery. A similar observation is valid for the score of inflammation (Table 3).

### 4.3. Donors’ Vascular Histopathologic Anomalies Are Recapitulated in the Rats’ Recipient Tissue

Healthy donor arterial CFF infiltration was neither associated to vascular changes nor to arterial walls pathologic remodeling as judged by the histological aspect and the wall-to-lumen morphometric measurements (Table 3 and Figure 2A). Worth mentioning however is the fact that arteriolar wall thickening was an invariant finding in those rats exposed to CFF derived from arteriosclerotic tissues of both donors 1 and 2. For both experiments, wall-to-lumen ratio measurements proved to be significantly enlarged as compared to healthy artery and normal saline treatments (Table 3). The arteriosclerotic artery homogenates affected granulation tissue neovessel morphology and induced pathologic changes similar to those found in the human donor tissues, in which arteriolar wall thickening was an invariant finding (Figure 2B).

The main donor’s changes reproduced included: (1) abnormal angiogenic response consisting in nests or nodules of malformed or incompletely organized vessels, containing tunnels with lumen-like spaces that as described for the donor are embedded within a matrix of collagen (Figure 2C), (2) media layer hyperplasia on the basis of collagen fibers accumulation with a notorious periarteriolar hypercellularity (Figure 2D), (3) non-thrombotic arteriolar luminal obliteration by protruding hypertrophic and fusiform-like cells that project into the lumen, eventually forming an encroaching mesh of trabecular structure (Figure 2E). As detected in the skeletal muscle samples of the donor patient (Figure 1G), the recipient rats exhibited in the leading edge of the *panniculus carnosum* the aberrant angiogenic response characterized by intramyofibrillar vascular structures including an endothelial collar (Figure 2F), often containing blood cells suggesting connection to the circulation. Of note, none of the changes described above were detected in the rats treated with the healthy donor/normal brachial artery, including the intramyofibrillar angiogenesis (Figure 2G).

### 4.4. Histochemical and Immunohistochemical Characterization of the Arteriolar Remodeling

Conventional histochemical techniques and a series of immunohistochemical reactions were conducted, concurrently analyzing the human donor pathologic tissue, the matched recipient rats and the healthy arterial donor recipient rats as controls. The formerly described arteriolar luminal encroachment in both the arteriosclerotic donors (Figure 3A) and the matched recipient rats (Figure 3B) proved to be positive to Mallory trichrome reaction. Furthermore, similar to the pathologic donor tissue (Figure 3C), the vascular walls of the recipient animals were positive to Congo red staining (Figure 3D). The control rats treated with the healthy arterial donor CFF were positive to collagen in a physiological pattern and negative to Congo red (Figure 3E,F).

Having observed that human pathologic changes were morphologically reproduced in the otherwise healthy recipient rats, we examined the immunohistochemical expression profile of critical markers in both the arteriosclerotic human donor, and recipient rats of abnormal and healthy samples. As an illustrative reference, Table 4 shows the contrasting expression between a normal artery and PAD-derived specimens in all the vascular biomarkers studied. Furthermore, the statistical analysis of the IRS indicated that the administration of arteriosclerotic-derived filtrates, significantly transformed the immunohistochemical landscape of the granulation tissue of recipient rats, as compared to concurrent control animals (Table 4).

Primarily, AGE and CD36 expression was studied considering their potential proximal role in vascular tissue damage. As shown in Figure 4A,B, AGE and CD36 were intensely and broadly expressed by the pathologic popliteal arteries (shown here samples from donor 1). However, these markers were not detected in the vascular structures or granulation tissue cells of the recipient rats (Figure 4C,D). Likewise, rats treated with the healthy donor/normal brachial artery CFF were also negative to AGE and CD36 expression (data not shown). Accordingly, the comparison of the IRS of these two markers for rats exposed to the pathologic and the healthy artery homogenates, showed no statistical differences (Table 4). Of note however, RAGE appeared intensely and broadly expressed by the pathologic arteries and the corresponding recipient rats. Rats’ granulation tissue vessels and resident cells displayed a large immunoreaction score as RAGE was conspicuously expressed (Figure 4E,F, respectively). In contrast, according to the scale used here, RAGE was not expressed in the group of control rats treated with the normal artery CFF (Figure 4G, and Table 4).

A similar pattern of intense immunoreaction was observed for NOX2/gp91phox within intramural infiltrating cells, possibly histiocytes in the human arteriosclerotic sample (Figure 4H). The exposure to the pathologic artery homogenate rendered a strong NOX2 expression score level given by the intense reaction of adventitial, fibroblasts-like cells, and granulation tissue perivascular infiltrates of mononuclear cells (Figure 4I). Contrariwise, NOX2 expression score was moderate in the granulation tissue of control rats treated with the healthy brachial artery CFF where few vascular cells were positive (Figure 4J and Table 4). TNF-α expression was found transmural and conspicuous in thickened arteriolar walls in the arteriosclerotic patients (Figure 4K) as in arteriolar walls and perivascular infiltrating cells in the corresponding recipient rats (Figure 4L). In these samples, TNF-α expression score significantly exceeded more than 3-fold the moderate reaction detected in granulation tissue inflammatory cells of rats exposed to the normal artery donor’s CFF (Figure 4M, Table 4).

The expression of an active/functional e-NOS isoform was found to be completely aborted by the endothelial collar and other cells within the human arteriosclerotic tissue sample (Figure 4N). In line with this, recipient rats of the pathologic artery CFF exhibited a weak e-NOS expression score being circumscribed to some perivascular infiltrating cells (Figure 4O). By the contrary, control rats treated with the normal brachial artery homogenate showed an intense and significantly superior immunoreaction score (Table 4), by different types of cells disseminated within the granulation tissue (Figure 4P). Finally, it was found that VEGF receptor-2 expression appeared blunted in the human arteriosclerotic tissue sample (Figure 4Q). Animals treated with this pathologic CFF (Figure 4R) mirrored this situation, showing a weak immunoreaction score. By the contrary, rats receiving the healthy donor brachial artery CFF exhibited a significantly larger immunolabeling score (Table 4). VEGF receptor-2 was expressed by cells of the endothelial collar as in areas of angiogenesis within the matrix of the granulation tissue (Figure 4S).

## 5. Discussion

With the caution attributable to the limitations of this work (incomplete mechanistic insights and limited number of experimental samples), it confirms and extends previous evidence on the hypothetical existence of a vascular memory. In this opportunity, the intralesional infiltration of a pathologic arterial tissue-derived CFF from non-diabetic subjects induced the reproduction of human donor’s representative arteriopathy markers in the rats’ granulation tissue arterioles within a period of days. The findings formally presented here completely reproduce those derived from pilot preliminary experiments, which were instrumental for definition of doses and administration schedule in the present protocol. Furthermore, the fact that this angiogenic pathologic remodeling was never detected in those rats treated with the brachial artery CFF from the healthy donor, suggests that the process described here is not a reactive response of the rats’ vascular walls versus the immunologic burden associated to human antigens.

As anticipated, the present experiment was prompted by previous evidences showing that diabetic wounds granulation tissue somehow “inherits” and recreates within a short period (days), vascular pathological changes (abnormal perivascular collagenization, thickening of *tunica media*, and ultimately a hyperplastic neointimal layer), analogous to diabetic cutaneous microangiopathy, which is considered of chronic progression [12]. The transference of a diabetic arteriosclerotic-tissue CFF to healthy rats’ wounds, translated in the phenotypical reproduction of the donor’s arteriolar thickening and the archetypical angiopathy histopathology [13]. Thus, the evidence presented here represent the third of a line of data supporting the notion of the existence of a kind of “pathologic vascular memory”, that leads to an abnormal angiogenic remodeling and that extends beyond the frontiers of the classic concepts of diabetic microangiopathy and metabolic memory.

Aside from the limited number of samples used and the donors’ individual biological variability, including the wall-to-lumen ratio in groups of control rats, it is notable that the two arteriosclerotic patients shared in common: (1) a severe endothelial dysfunction as judged by the measurements of nitrite/nitrate ratio level relative to the healthy artery homogenate [41,42], and (2) a dramatic reduction in the arterial content of sirtuin-1 levels. Recent studies have shown that sirtuin-1 plays a unique role in vasoprotection and that sirtuin-1 and e-NOS regulate each other synergistically through positive feedback mechanisms for the maintenance of endothelial physiology and vascular aging [43,44]. Thus, although the human arteriosclerosis pathophysiology and the order of its hierarchic events remains elusive [45,46], the findings presented here suggest the existence of soluble messengers or signalers derived from the donors’ pathologic arteries which in common drive to the recreation of the human condition in a distant animal species.

The impact of the infiltration with the arteriosclerotic material was not limited to morphological remodeling but also to functional implications. Granulation tissue arterioles of the recipient rats recreated the donor’s luminal occlusion by the encroachment of fibroblast-like cells, apparently emerging from the endothelial collar and associated to collagen bundles positive to Mallory staining. Recent studies characterizing the non-thrombotic luminal narrowing of small arterioles in CLI patients, describe the phenotypic shift of cuboidal endothelial cells to interstitial fibrogenic cells, that displayed molecular attributes of partial endothelial-to-mesenchymal transition [47].

Based on the histological resemblance among the description of the human endothelial-based microvascular stenosis by phenotypic reprograming [47], the donor patients’ fibrotic remodeling, and the rats’ arteriolar fibrogenic luminal encroachment; we hypothesize that patients-derived CFF contains signaling messengers involved in the process of endothelium-to-fibroblast reprograming. As a matter of fact, our descriptive observation of a reduced VCAM1 content in the human pathologic arteries, is considered a *bona fide* marker of endothelial-to-mesenchymal transition [48]. The hypothetical involvement of cells and tissue reprograming processes, via the participation of signaling messengers, is further supported by our findings of “aberrant angiogenesis” in the skeletal muscle of both the arteriosclerotic donor and the recipient rats. Interestingly, this type of “aberrant angiogenesis” in which areas of muscle myofibrils are replaced by vascular-like channels that may contain an endothelial collar, and represent the replacement of one tissue lineage for an unrelated one, has been solely attributed to long term exposure (12 months) of deregulated concentrations of VEGF-A_165_ [49]. Of note, all the tissue remodeling events described in our recipient rats evolved in a period of 7 days whereas these events in humans are conceptually considered as of chronic evolvement. Examination of our endothelial-to-mesenchymal transition hypothesis is warranted in future experiments. The potential identification of the proximal drivers of this process may have important prophylactic and therapeutic implications.

The comparative immunohistochemistry experiments showed that rats treated with the pathologic donor tissue CFF, adopted the donors’ abnormal immunohistological picture, given by a statistically significant elevation of the immunoexpression score for RAGE, NOX2, and TNF-α; versus a meager expression of functional e-NOS and VEGF receptor-2, in reference to the concurrent control rats. It is interesting however that the “acquisition” of these morphological and expression phenotypes by the pathologic donor-recipient rats, was not associated to rats’ tissue AGE accumulation and/or CD36 expression. Contemporary hypotheses concerning the molecular pathology of the atherogenic development include the interconnected involvement of AGE toxicity [50,51], and the classic atheroinflammatory scavenger receptor CD36 [52,53]. Thus, further studies will be addressed to identify the drivers promoting these arteriolar pathologic remodeling. The observation of a coexisting Congo red staining in the granulation tissue of the recipient rats as in the vessels of the original arterial pathologic material is intriguing. Whether this finding represents the passive transference of amyloid, a similar reactant material, or a proximal signaling factor promoting acute vascular amyloid accumulation remains elusive.

As mentioned above, granulation tissue arteriolar wall thickening was an invariant observation in both experiments, irrespective to the biochemical differences detected in the pathologic donors’ homogenates. We deem that the factor(s) involved in the phenomenon described here may not be a single molecule, but a combination of signaling agents, including chronic hypoxic tissue signalers and/or exosomes-containing epigenetic effectors. Exosomes, that comprise a constellation of subcellular fragments appear to play important roles in cell-to-cell communication, horizontal gene transfer, immune modulation, and in the development of diabetes and its complications [54]. The harmful participation of pathological exosomes in cardiovascular remodeling has been carefully reviewed [55]. Furthermore, converging evidence [56,57] documents that those vesicles messengers are involved in endothelial-mesenchymal transition leading to conversion of endothelial cells to fibroblasts which ultimately promote luminal obliteration with fibrotic material as that described in our rats.

It is our hypothesis that the onset of this non-thrombotic, arterial wall remodeling in the human subject has as a proximal trigger an abnormal epigenetic landscape reprogramming, and that this process, is encrypted through chromatin and non-chromatin modifications which are able to withstand two major events in the self-human host: (1) Internal dissemination of the disease through the release of epigenetic drivers, and (2) Its perpetuation and irreversibility. The latter is a representation of the existence of a “pathologic vascular memory”, whose epigenetic message seems to be soluble and transferable, as it may faithfully drive the reproduction of the donor’s histopathologic traits in a normal animal. We also deem that the “phenotypic dominance” imposed by these pathologic donor-derived drivers, over the rats’ normal angiogenic process, is a mere reflection of the existence of such “memory”. Conclusively, we must demonstrate that rat vascular tissues are “contaminated” with a “vascular pathologic memory” after being exposed to human epigenetic drivers of PAD contained in the homogenates. To address this critical question, “second generation” studies are planned in which rat wounds tissues exposed to human PAD homogenate will be used as the primary source of homogenates, and if its subsequent administration to other recipient animals drives the reproduction of the original human arterial pathology.

The futuristic implications of these combined pieces of evidence justify additional efforts to confirm evidence reproducibility, as this protocol was based on the tissues collected from only two PAD/CLI donors. Taken together, our studies suggest that in addition to the well-established risk factors for non-diabetic PAD, there is a sort of pathologic vascular memory represented by soluble messengers that can spread and perpetuate the vascular disease. These “drivers” can be transferred without inter-species barriers from humans to normal healthy animals, where they impose their vascular pathologic program. This study supports and extends our previous observations on the existence of a pathologic vascular tissue memory and its transferable character.

## Figures and Tables

**Figure 1 ijms-23-01511-f001:**
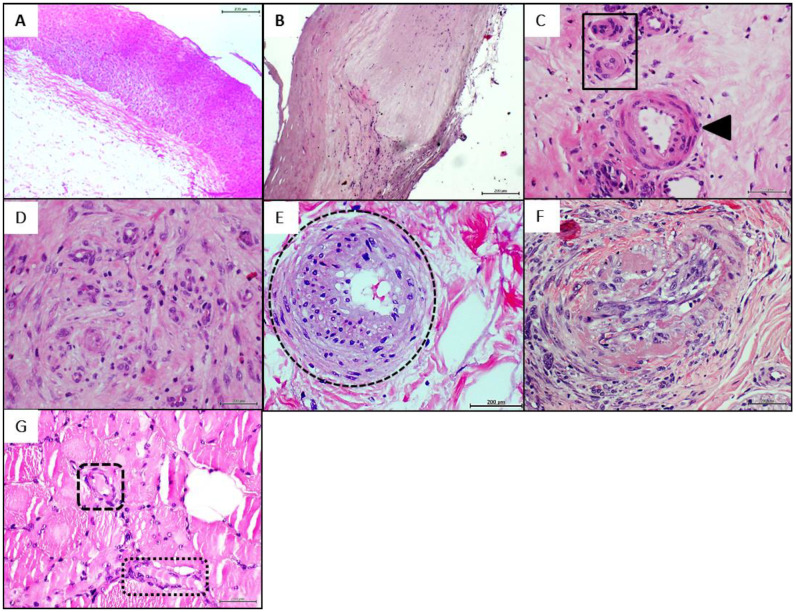
Histological characterization of human donors’ tissues by hematoxylin/eosin staining. (**A**): Fragment of the healthy donor brachial artery showing transmural preservation of the vascular wall. Magnification 10×. (**B**): Arteriosclerotic donor’s tissue used to prepare the homogenates (popliteal wall) showing fragmentation of the intima elastica, fibrotic media layer thickening and intima invasion. Magnification 10×. (**C**): Illustrative of abnormal vessels growth in foci of compensatory angiogenesis. The arrowhead indicates an arteriole with an expanding media layer and hypertrophied endothelial nuclei. The square shows two incipient small arterioles that appear already occluded. Magnification 40×. (**D**): Angiogenesis was frequently found in cluster-like structure with internal collagen strands, containing incipient or incompletely formed neovessels and cellular disorganization. Magnification 20×. (**E**): The circle encompasses an arteriole with an ongoing process of intimal hyperplasia, subendothelial infiltration, luminal narrowing and concentric expansion of the media layer with large basophilic nuclei. Magnification 40×. (**F**): Image representative of a preexisting arteriole showing degenerative wall changes including accumulation of hyaline material and internal *tunica elastica* fragmentation. Some endothelial cells are hypertrophied and intermixed with fibroblast-like cells that appear to emerge from the endothelial collar and make up a mesh-like structure that encroaches into the lumen. Magnification 40×. (**G**): The squares delimit the microscopic image of “aberrant angiogenesis” in the skeletal muscle of the arteriosclerotic donor patient limb. The intramyofibrillar vessels exhibit an endothelial collar and luminal content. Magnification 40×. For all, the scale bar was 200 µm.

**Figure 2 ijms-23-01511-f002:**
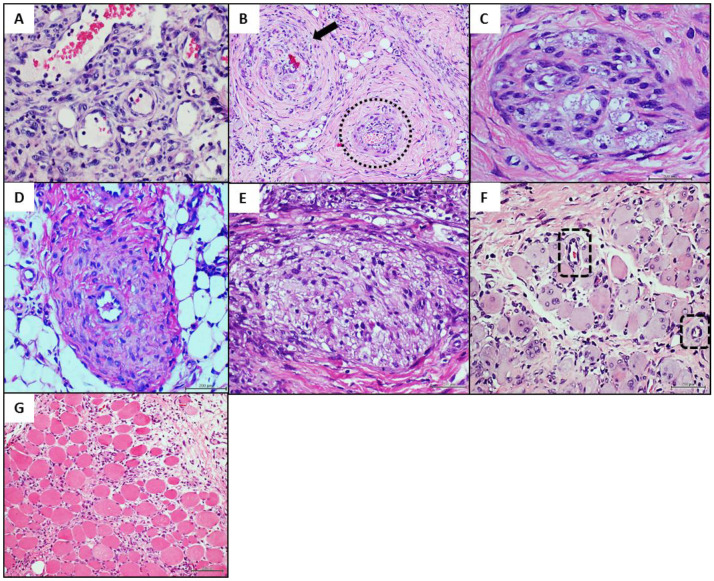
Histological characterization of recipient rats’ granulation tissue by hematoxylin/eosin staining. (**A**): Representative of normal granulation tissue vessels in control rats infiltrated with healthy donor/normal brachial artery CFF. An abundant number of morphologically normal neovessels with patent lumen are shown. Granulation tissue matrix is infiltrated by round cells and other with fibroblastic-like phenotype. Magnification 40×. (**B**): Represents the granulation tissue vascular response to arteriosclerotic material-derived CFF in which a neoformed arteriole exhibit intense wall thickening (arrow), abnormal cellular infiltration and disorganization. There is an exaggerated accumulation of circular and concentric collagen bundles. The lumen is almost completely obliterated. The doted circle shows another arteriole affected by intimal expansion associated to subintimal cellular infiltration. Magnification 20×. (**C**): A node of abnormal and incomplete angiogenesis in rats infiltrated with the arteriosclerotic material-derived CFF is shown, having tunnel-like structures often with scarce or no endothelial cells lining. Other large basophilic nuclei are also evident. There is an external ring of cells with endothelial and fibroblastic-like phenotypes. Magnification 40×. (**D**): A granulation tissue arteriole exhibiting a significant media layer hyperplasia due to collagen accumulation with a notorious hypercellularity. Magnification 40×. (**E**): The image shows the remnant of an arteriole with complete luminal invasion by protruding hypertrophic and fusiform-like cells that appear to emerge from the endothelial side and project into the lumen, eventually forming a trabecular structure. The luminal mesh is infiltrated by round basophilic nuclei-suggestive of lymphocytes and other of fibroblastic aspect (compare to Figure 1F). Magnification 40×. (**F**): The doted squares show the presence of vascular structures with or without endothelial collars, organized within the myofibrillar parenchyma of the *panniculus carnosum* layer in rats treated with the pathologic donors’ filtrates. This represents an aberrant angiogenic response of the recipient rats to arteriosclerotic material-derived CFF. Magnification 20×. (**G**): Cross section of the *panniculus carnosum* layer leading edge, showing the normal integrity of the myofibrils in control rats treated with healthy donor/normal brachial artery CFF. Magnification 20×. For all images, the scale bar represents 200 µm.

**Figure 3 ijms-23-01511-f003:**
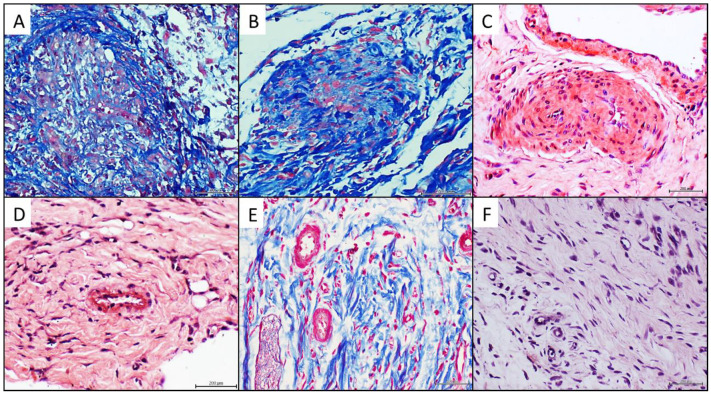
Histochemical analysis of pathologic donor tissue and recipient rats’ granulation tissue. (**A**) Arteriosclerotic donor-derived sample showing an arteriole positive to Mallory staining to demonstrate collagen material (pale and dark blue). Collagen strands seem to emerge luminal side of the vessel wall and invade into the luminal aspect. (**B**) Representative of the process of collagenous luminal invasion in an arteriole in the granulation tissue of a rat treated with the arteriosclerotic donor homogenate. An exaggerated collagen accumulation (dark blue) is observed the media layer sector. Pale blue strands also invade and occupy the lumen. (**C**) Thickened neoformed arterioles in the arteriosclerotic donor showing positive staining to Congo Red, indicating amyloid type-material accumulation. (**D**) Similar positive reaction to Congo Red is identified in the endothelial collar of an arteriole in a rat treated with the human arteriosclerotic tissue homogenate. (**E**) Mallory reaction in the granulation tissue of rats treated with the healthy donor/brachial artery homogenate. Collagen positive strands are normally aligned in the wound matrix, and that there is no angiocentric accumulation. The luminal aspect of the vascular walls is well-delimited and there is no collagen invasion to the lumen. (**F**) denotes that the administration of the healthy donor normal brachial artery was not associated to Congo Red positive staining. Magnification 40×. For all images, the scale bar represents 200 µm.

**Figure 4 ijms-23-01511-f004:**
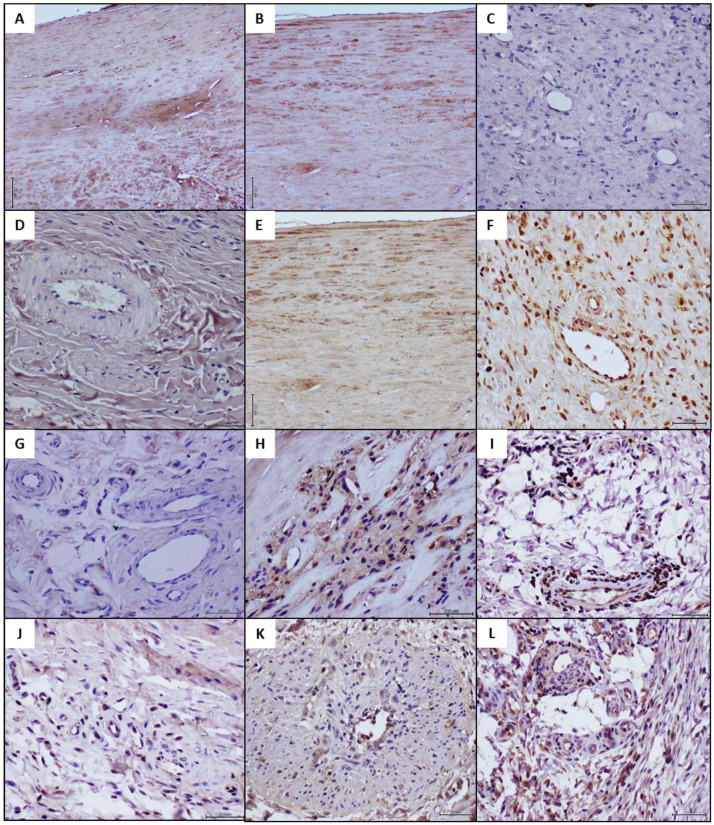
Immunohistological characterization of pathologic donor tissue and recipient rats’ granulation tissue with vascular biology-relevant antibodies. (**A**): Fragment of the arteriosclerotic popliteal artery wall showing positive signal to AGE accumulation in a transmural pattern. Similarly, the labeling to CD36 as illustrated in (**B**) is noticeable and almost transmural. (**C**,**D**) demonstrate no expression by granulation tissue structures of AGE and CD36 respectively, in those rats receiving the arteriosclerotic popliteal artery homogenate, which contrasts to the expression pattern found in the donor’s artery. Magnification 20×. (**E**) Demonstrates that RAGE is intensely expressed by the arteriosclerotic popliteal arterial tissue. Magnification 20×. (**F**): Recipient rats of the arteriosclerotic material homogenate evidently express RAGE in granulation tissue cells as in the vascular endothelial collar. Magnification 20×. By the contrary (**G**), no signal of RAGE expression was revealed in the granulation tissue of rats receiving the homogenate of the healthy donor brachial artery (control). Magnification 40×. (**H**): NADPH oxidase (NOX) expression is detected within an area of intramural infiltration of inflammatory cells in the arteriosclerotic popliteal artery of the human donor. Rats treated with this pathologic material homogenate congruently expressed strong NOX signal in perivascular infiltrated inflammatory cells as shown in (**I**), whereas the control rats treated with the normal brachial artery moderately expressed NOX in a limited number of cells adjacent to vascular walls (**J**). Magnification 20×. (**K**) demonstrates that TNF-α is intensely expressed by arterioles adjacent to the pathologic popliteal artery in the arteriosclerotic donor. The expression involves the thickened and hypercellular wall and the endothelial collar. Smaller satellite vessels also express the cytokine. Broad and strong TNF-α expression is also detected in granulation tissue cells and vascular structures of rats treated with the arteriosclerotic material homogenate (**L**). Of note, however, is that TNF-α expression is moderate to weak in limited cells within the granulation tissue of control rats treated with the normal brachial artery as shown in (**M**). The use of an antibody reactive to an active isoform of endothelial nitric oxide synthase (e-NOS) suggested that arteriolar pathologic remodeling is associated to the abolition of the enzyme expression as seen in (**N**), representative of the arteriosclerotic donor tissue. Interestingly, this expression deficit was also detected in the granulation tissue vessels of rats treated with the pathologic material homogenate. (**O**) illustrates that e-NOS expression is fairly marginal (moderate scoring, Table 4) in vascular cells as compared to the strong and diffuse expression found in granulation tissue samples of control rats (**P**). Magnification 40×. VEGF receptor-2 expression appeared to be reduced in the limb tissues adjacent to the popliteal artery of the arteriosclerotic donor (**Q**). Rats treated with this pathologic material homogenate also showed a clear reduction of VEGF receptor-2 expression in vascular walls (**R**) and adjacent cells as compared to control rats (**S**) treated with the healthy donor brachial artery. Magnification 40×. For all images, the scale bar represents 200 µm.

**Table 1 ijms-23-01511-t001:** Immunohistochemistry using vascular biology relevant markers.

Antibody	Biological Implication	Reference
Anti-TNF-α (Ab6671)	TNF-α. Pro-inflammatory cytokine involved in the pathogenesis of arteriosclerosis	[28,29]
Anti-NOX2/gp91phox (Ab31092)	NADPH oxidase (NOX) activity producing reactive oxygen species, involved in arteriosclerosis development	[29,30]
Anti-RAGE (Ab16329)	RAGE is activated in hyperglycemia and hyperlipidemia, inducing pro-inflammatory responses and oxidative stress in a variety of vascular diseases	[29,31,32]
Anti-AGE (Ab23722)	Evidences indicate that interaction of AGE with their receptor (RAGE) elicits oxidative stress generation and as a result evokes proliferative, inflammatory, thrombotic, and fibrotic reactions in a variety of cells leading to vascular pathology	[33,34]
Anti-CD36 (Ab78054)	CD36 is a class B scavenger receptor and membrane glycoprotein playing a critical role in atherosclerotic onset and development	[35,36]
Anti-endothelial NOS phosphorylated in serine-1177 (Ab75639)	e-NOS plays a major role in angiogenesis, vascular homeostasis and protection. Failure of e-NOS function has been implicated in hypertension, thrombosis, atherogenesis, and abnormal vascular reactivity	[37,38]
Anti-VEGF receptor-2 (Ab2349)	VEGFR-2 is the major receptor for VEGF, expressed in vascular endothelial cells. It is implicated in angiogenesis, vascular cells survival, migration, and proliferation. Responsible for the conservation of a normal vascular anatomy	[39,40]

**Table 2 ijms-23-01511-t002:** Biochemical characterization of the experimental samples.

Sample	IL-6/mg Prot. (pg/mg)	IL-1β/mg Prot. (pg/mg)	AGE/mg Prot. (ng/mg)	MDA/mg Prot. (nmol/mg)	VCAM1 (ng/mL)	Sirtuin-1 (ng/mL)	Nitrite/Nitrate Ratio
Donor 1 Peripheral arterial disease	65.24	33.99	5.635	0.313	2.15	N.D. < 0.29	0.99
Donor 2 Peripheral arterial disease	3.99	17.53	1.022	0.464	2.15	0.78	0.56
Healthy donor/Normal artery	10.43	16.97	0.676	0.601	8.86	3.26	2.81

N.D.—not detected (sirtuin-1 levels below 0.29 ng/mL). AGE: advanced glycation end products; IL-1β: interleukin-1 beta; IL-6: interleukin-6; MDA: malondialdehyde; prot.: protein; VCAM1: vascular cells adhesion molecule 1.

**Table 3 ijms-23-01511-t003:** Effect of peripheral artery disease arterial tissue homogenates on healthy rats’ granulation tissue histological appearance and wound healing rate.

	Wound Closure (%)	Arterial Thickening (Wall to Lumen Ratio)	Nerve Fibres Damaged (%)	Inflammation
Experiment 1				
PAD artery 1	47.80 ± 6.40 (a)	1.78 ± 0.39 (a)	58.15 ± 14.87 (a)	7.87 ± 1.83 (a)
Healthy artery	53.78 ± 13.30 (a)	0.37 ± 0.12 (b)	53.47 ± 29.35 (a)	5.57 ± 2.39 (b)
Saline	52.45 ± 9.55 (a)	0.34 ± 0.11 (b)	40.80 ± 15.94 (b)	2.03 ± 0.17 (c)
Experiment 2				
PAD artery 2	68.08 ± 10.03 (b)	1.15 ± 0.29 (a)	73.98 ± 13.14 (a)	2.86 ± 1.25 (a)
Healthy artery	79.65 ± 9.12 (a)	0.66 ± 0.18 (b)	53.51 ± 24.60 (a,b)	2.38 ± 1.12 (a)
Saline	74.12 ± 7.65 (a,b)	0.62 ± 0.35 (b)	41.72 ± 19.60 (b)	1.09 ± 0.27 (b)

Data are presented as mean ± SD. Statistical analyses were performed through one way ANOVA followed by Holm-Sidak’s multiple comparisons test, or by the Kruskal-Wallis test followed by Dunn’s multiple comparisons test, for data with or without normal distribution, respectively. Different letters indicate statistically significant differences among experimental groups (at least *p* ≤ 0.05). PAD: peripheral arterial disease.

**Table 4 ijms-23-01511-t004:** Immunoreaction score (IRS) calculated for human arterial specimens and rats’ granulation tissue treated with human arteries-derived homogenates.

Markers	Human Donor Samples		Recipient Rats		
	HAD	PAD	HAD	PAD	*p* Value
AGE	0.00	7.75	0.00 ± 0.00	0.26 ± 0.46	0.6016
CD36	1.50	4.88	0.59 ± 0.60	0.31 ± 0.46	0.2948
RAGE	0.00	12.25	0.41 ± 0.50	16.35 ± 3.58	0.0002
NOX2	0.25	9.70	5.46 ± 0.80	11.68 ± 2.11	<0.0001
TNF-α	0.50	13.10	4.03 ± 1.84	13.08 ± 2.14	0.0003
e-NOS	8.27	0.88	14.21 ± 1.46	4.44 ± 1.15	0.0002
VEGFR-2	11.50	0.93	17.60 ± 2.94	3.82 ± 1.53	<0.0001

IRS score calculation was done as described in Materials and Methods. Data are expressed as absolute values for human donor samples (1 healthy and two arteriosclerotic donors) and as mean ± SD of two independent score rounds of 7–8 animals in the case of recipient rats. Statistical analyses for recipient rats were performed using unpaired *t* test (VEGFR-2, NOX2, CD36), Mann Whitney test (TNF-α, RAGE, e-NOS), or Wilcoxon signed rank test (AGE). AGE: advanced glycation end products; e-NOS: endothelial nitric oxide synthase; HAD: healthy artery donor; NOX2: NADPH oxidase; PAD: peripheral artery disease; RAGE: Receptor for AGE; TNF-α: tumor necrosis factor-alpha; VEGFR2: vascular endothelial growth factor receptor 2.

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
