# Peer review of "Intralesional Infiltrations of Arteriosclerotic Tissue Cells-Free Filtrate Reproduce Vascular Pathology in Healthy Recipient Rats"

_ijms, 2022, doi:10.3390/ijms23031511_

Round 1

Reviewer 1 Report

Manuscript by Berlanga-Acosta et al describes the effect of cells-free filtrates (CFF) obtained from arteriosclerotic human tissue on the vascular pathology of wound healing in healthy rats in which a wound is induced. The authors present data to demonstrate that vasculature of the healing wound in rats which are subjected to arteriosclerotic CFF exhibit characteristics that are similar to those observed in the donors’ arteriosclerotic vessels. The characteristics examined include vessels morphology such as vascular thickening and abnormal angiogenesis, as well as expression of, inflammatory, oxidative, and endothelial specific markers, including TNFa, RAGE, VEGFR2 and eNOS expression. The study is conceptually interesting and potentially impactful. However, there is a main concern regarding the sample size and quantification, which makes it difficult to be unequivocally convinced of the conclusion arrived by the authors.

Major concerns

- As authors have also noted the small sample size (two arteriosclerotic and one healthy samples) are the major limitation. However, this is confounded by significant inter experimental variability.

For instance, in Table 3. There is a significant difference in histological appearances and wound healing rate measurements when comparing the controls in two experiments. i.e Arterial thickening is 0.37 in healthy artery in experiment 1 and 0.66 in experiment 2. This is similar also for saline treatment between the two experiments. This inter experimental variability makes it difficult to conclude that differences observed when comparing only two PAD to one healthy artery are above inter-experimental variations.

- The other related concern is the representative IHC data presentation without any form of quantification. IHC staining is not a quantitative measurement and to obtain some measure of quantification, at minimum the IHC could be analyzed by scoring staining in multiple samples and multiple field of views per sample. Since the authors stated that 8 rats were used per experimental groups this should provide an opportunity for such quantifications. Additionally, why not perform Western analyses on rat tissue samples for the target proteins described in Fig. 4 to obtain quantitative data?

Minor concerns

- In the list of antibodies used Ab75369 is stated as Anti-endothelial NOS phosphorylated in serine 1177. However, this antibody appears to be Rhodopsin antibody. Also, the TNF-a antibody (Ab6671) is not reported as being reactive to rat. Have the authors confirmed the specificity of the rat TNFa detection by this antibody?

- Some editorial modifications to clarify certain statements would be helpful.

For instance in page 2 second paragraph the meaning of the following statement is not very clear: “Thus, although granulation tissue (GT) is considered a “de novo” and transient “welding tissue,” it hosts endothelial cells that may be epigenetically instructed for the “acute inheritance” and faithful reproduction of diabetic microangiopathy traits [12]. 

Similarly in page 3 the following sentence at the bottom of the page needs editing: “…….which clarified any confounding effect associated to the rats’ tissue reaction before human xeno-geneic material. 

Similarly Discussion section can benefit from editorial modifications for clarification.

Author Response

Thank you very much for the excellent criticism. We will address each and every one of the concerns.

Reviewer 2 Report

The subject of the article is attractive, however the number of Peripheral arterial disease are too small, either increases the number of patients, or remove the article and resubmit when you have more data. This article is too preliminary.

Author Response

Thank you very much for the criticism of the manuscript. Yes, it is correct that the number of patients is small and it is somehow precipitate to draw definitive conclusions. However, given the logistical hurdles imposed by COVID-19 outbreak in our country we do not see a clear and immediate horizon to obtain other surgical samples of amputee subjects suffering of chronic limb ischemia, and therefore substantiate the evidences. I wonder if you deem adequate to add the statement "a preliminary evidence" in the title of the manuscript as an alternative to save this conundrum. In addition we can further clarify it in the Discussion when describing the limitations of the work. Thank you for your attention.

Round 2

Reviewer 1 Report

The  concerns raised by this reviewer were addressed adequately and the manuscript was significantly strengthened.

A minor point regarding clarification of some sentences throughout the manuscript remains. It will also help to describe how the authors connect  the observations with their hypothesis related to "vascular tissue memory"? The results are consistent with the authors notion of "vascular disease drivers", however it is not clear to this reviewer how this relates to "memory " since this implies that the rat vascular tissues have had a potentially epigenetic  "memory" of being exposed to PAD. 

Author Response

Havana, November 21, 2021.

Dear Reviewer.

Thank you very much for this wise and rewarding question/comment. I mean rewarding as it reflects you have penetrated to the essence of the work. Please find our hypothesis below, and a new paragraph inserted in Discussion on page 22.

Donor’s vascular pathologic tissue of human origin is associated to the onset of a privative epigenetic landscape reprogramming, encrypted through chromatin and non-chromatin modifications, which to our notion, mediate two major aspects in the self-human host: (1) Internal dissemination of the disease through the release of epigenetic drivers, and, (2) Perpetuation or irreversibility of the process. The faithful recapitulation of the histopathologic traits of the donor’s vascular pathology - in a normal animal, is a reflection of the existence of such “pathologic memory” which may be passively transferred and reproduced in other species. Thus we use the concept of “memory” to describe the precise reproduction of the donor’s pathology in the host model. I also hypothesize that there is a certain degree of dominance of the pathologic phenotype versus the normal host’s tissues. Please note that the present study as the previous one using diabetics-derived tissues, (Front. Clin. Diabetes Healthc. 2:617741. doi: 10.3389/fcdhc.2021.617741) are based on the local delivery of the normal or pathologic tissues homogenates into the matrix of healing wounds – this means that it is likely that the donor’ pathologic drivers may disrupt the host’s normal angiogenesis process- typical of the healing process. Thus far it seems there is a type of “dominance” in which the abnormal overrules the normal. We have done experiments with intact rats, subcutaneously injecting the normal and pathologic homogenates in specific sites, and we are not sure to have successfully reproduced the human condition. The tissue damages associated to the daily local injections trauma- are somehow eclipsing to draw a firm conclusion. As you have clearly remarked - we have to show that the rats’ tissues are potentially driven by an epigenetic “memory"- being as a sort of a heritage after be being exposed to the PAD-derived homogenate. In addition to molecular genetic screenings, we are planning “second generation” experiments in which the rats wounds tissues exposed to human PAD homogenate, will be used as the primary source of homogenates, and if its subsequent administration to other recipient animals, drives the reproduction of the original human arterial pathology. Finally, I am delighted to share with you that we have succeeded in inducing a set of malignant tumors in nude mice in just 6-to-12 weeks by daily injecting homogenates derived from both breast ductal adenocarcinomas and an intrathoracic anaplastic sarcoma.

Thank you for your constructive criticism.

Reviewer 2 Report

This article is too preliminary, only have 2 donors.

Author Response

Dear Reviewer.

Thank you very much for your comment. You are correct, the article is preliminary and we have acknowledged in Discussion. We respectfully believe however that still, there are converging and reproducible findings that deserve to be anticipated.  We beg your consideration. Thank you very much.

Round 3

Reviewer 2 Report

This article was revised appropriately.

I recommend accept